# Learning to Understand Videos From Encoded Bytes

**AJ Piergiovanni, Ganesh Mallya, Dahun Kim, Anelia Angelova**
Google DeepMind
{ajpiergi,ganeshmallya,mcahny,anelia}@google.com

Reviewed on OpenReview: https://openreview.net/forum?id=psLcyKiuCp

## Abstract

We present an approach to understand video from encoded bytes, e.g., mp4s. These compressed videos are 99% smaller than the RGB pixel representations which are currently commonly used for video understanding. Encoded videos are able to compress the pixels by taking advantage of the redundant information across the frames using special encoding, such as key frames and motion residuals to handle this. However, standard video understanding models do not take advantage of this significant compression already available for each video, and instead either heavily subsample the frames or only work on short segments of the video. Here, we present an approach to understanding video from encoded bytes directly. We note that simply applying existing models, e.g., Transformers or State-Space models, to video byte sequences does not work, both due to difficulty in handling very long video byte sequences and easy overfitting. To address these challenges, we design a State-Space model with sequence parallelism to handle very long byte sequences, reaching **15 million tokens** in training, and enabling further gains by even longer token lengths in inference. We also propose a multilevel SSM activation fusion that reduces sequence length, which we find also benefits video understanding. We evaluate on six video understanding benchmarks including long, high-fps and video + audio understanding tasks and demonstrate competitive performance, illustrating, for the first time, the feasibility of learning from compressed video byte representations.

## 1 Introduction

Video understanding is an important problem in computer vision. Unlike images, videos contain multiple frames. In traditional settings, these frames are treated as a sequence of images (Simonyan & Zisserman, 2014; Carreira & Zisserman, 2017; Yue-Hei Ng et al., 2015; Tran et al., 2015), which greatly increases the compute costs and memory requirements and makes it hard to scale to longer and longer videos. Here, we propose an alternative approach which instead understands videos as encoded bytes, e.g., mp4 byte streams. The main advantage is the significant memory savings in processing compressed video, since the compression codecs take advantage of redundant pixels in consecutive frames, they are greatly able to reduce the size of a video. Moreover, the video byte streams are naturally suited for sequential models, as they are designed for video streaming and playback, where the decoders reconstruct the sequential frames from the bytes. And further, byte streams do not contain strong inductive biases, and so do not require operations like convolutions as in ViTs (Dosovitskiy et al., 2020). While byte-based representations are highly compressed, both pixel and byte representations contain the same information, and both have a complex, inconsistent, non-linear mapping between the inputs and semantic understanding, thus it is theoretically possible to learn any video understanding task from either input, which we explore here.

As a motivation, using the standard video representation of float16 $[F \times H \times W \times 3]$, and assuming 30fps and 480 resolution, one would need roughly 40 megabytes of memory per second of video for storing just the pixels (not including the model weights or intermediate activations). For a 10 minute video at 30 FPS and 480 resolution, this would use roughly 25GB of memory just for the pixel inputs. However, that same video

Table 1: Our approach is able to handle very long byte sequences from raw compressed video formats, supporting up to **15 Million byte tokens** during training, leading to significant performance improvements.

| Sequence Length | ActivityNet-QA Acc (%) |
|---|---|
| 250,000 | 42.5 |
| 2 Million | 54.2 |
| 15 Million | **56.3** |

Table 2: Testing on longer sequence lengths **(20 Million bytes, which corresponds to ∼13 minutes of video)** than training (15M bytes) yields further gains. While the proposed approach can theoretically support unlimited tokens during inference, the effective context window in practice is likely bounded by factors, such as, recency bias, signal attenuation and others.

| Sequence Length | ActivityNet-QA Acc (%) |
|---|---|
| 15 Million | 56.3 |
| 20 Million | **57.1** |

encoded with a bitrate of 2Mbps (YouTube's recommended compression rate for 480p videos), would use only about 150MB for the same 10-minute video, which, compared to 25GB, yields a significant reduction in filesize, i.e., about 99% smaller. If we are able to perform video understanding on compressed byte inputs, this will allow for great compute savings and enable scaling to long videos.

Further, existing specialized video models either focus on modeling sparse temporal relationships on image-based features (e.g., (Yue-Hei Ng et al., 2015; Piergiovanni et al., 2017; Chen et al., 2023; Lin et al., 2025)) or use tube-based features on short segments of video (Arnab et al., 2021; Piergiovanni et al., 2023) or apply segment-based pooling (Shou et al., 2016). These prior works perform well on video, but are essentially leveraging existing image-based models for short or sparsely sampled video segments. Instead here, based on the observation that compressed videos contain all the information needed to reconstruct the pixels, but with much less redundant information, we design an approach to directly learn specifically from video bytes. We note that many video datasets contain short clips, and datasets focused on long videos (LVBench (Wang et al., 2024a), MLVU (Zhou et al., 2024), VUE-TR (Team et al., 2025b), Neptune (Nagrani et al., 2024), etc.) are either being used for evaluation-only or are trained on relatively sparse sampling of the video frames due to memory constraints. For example, recent works, such as InternVideo2 (Wang et al., 2024b) and VIDI (Team et al., 2025b) only train on short segments with 8 frames and 1 fps up to 120 frames, respectively, and Hour-LLaVA (Lin et al., 2025) very sparsely samples video tokens from a frozen image encoder.

Training on video bytes allows using all the frames with much less memory, however, as we show in the experiments, this is a non-trivial task due to a few issues. First, encoded video bytes are very long sequences. Instead of a video being a $[F \times H \times W \times 3]$ that can be further compressed with spatial and temporal pooling, we have an input of $[L \times D]$, where $L$ is the sequences length and $D$ is the embedding size of the model. For video bytes, this results in sequences with millions of tokens, which is extremely long even compared to modern LLMs (e.g., LLAMA 3.1 (Grattafiori et al., 2024) supports 128,000 tokens in inference, Gemma (Team et al., 2025a) was trained with an 8k sequence length). This presents a real problem, as this becomes a long-sequence length learning problem. Second, we find that Transformers are not the best suited model for this task, due to the long sequences and poor scaling of self-attention. Finally, understanding the encoded representation is far more challenging than understanding pixels, as the representation is much more compressed. Thus, taking an existing LLM model and directly training it on encoded video bytes does not perform well at all. We propose a method to learn from encoded video bytes that works on very long sequences, which are of different structure than text inputs, and we develop a multilevel, sequentially-parallel modeling of video bytes and find it is well suited for this challenging task. We make several key contributions to enable video understanding from bytes:

- We present the first approach to understand video from raw, encoded bytes, circumventing the decoding process which increases the video volume processed by over 100x, and leveraging the highly compressed video inputs which are ubiquitous video representations.

- We propose an efficient parallelization and gradient accumulation method, with a correction and propagation of the state, based on the sequence parallellism paradigm Gu & Dao (2024). This

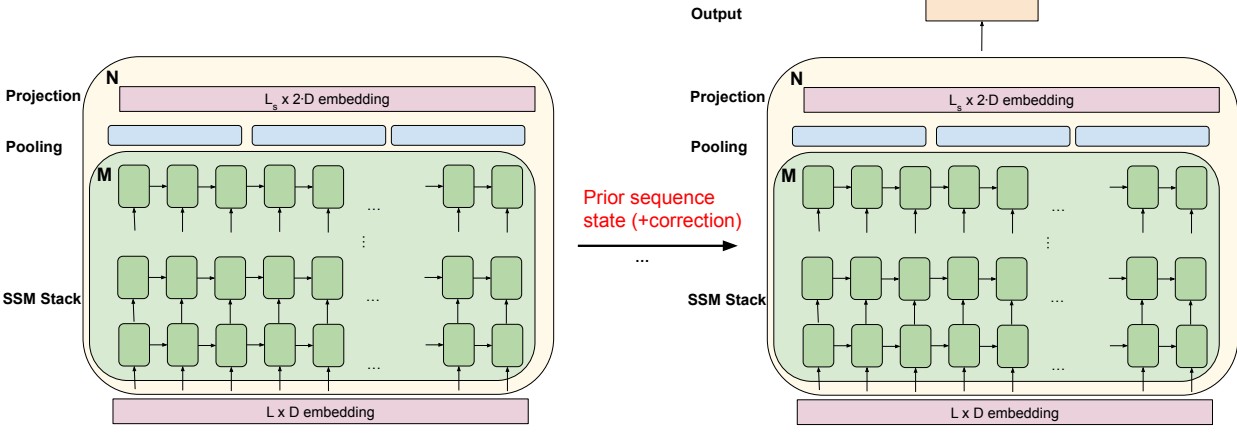

Figure 1: Model outline. A sequentially parallel model (Sec. 3.1), here validated with a Mamba SSM, allows for parallel processing and accumulates and corrects the model state, ensuring the retention of information beyond what can fit in device memory. For further scaling, the raw bytes are embedded, then processed by SSM layers (here $M$) in a multilevel module (Sec. 3), which is repeated $N$ times. This enables efficient paralelization of very long sequences.

> enables training on multi-device and on extremely long bytes sequences, e.g., **15 million** ( Tables 1, 2), and even longer sequence lengths during inference.

- To further scale, a 'multilevel' SSM efficiently accumulates SSM activations and, together with the sequence parallelism, enables scaling to very long sequences during both training and inference. This model greatly outperforms standard SSMs and Transformers when applied to encoded video bytes (Table 9).

We note that here we do not use any prior knowledge about the structure of the bytes, e.g., information about the codec, which makes the task significantly harder, but also more general. We show that data-augmentation addresses the issue, noting that injecting additional information about the sequence structure can be further leveraged in future work.

## 2    Learning from Raw Video Bytes

Our approach takes a sequence of raw video bytes as input, rather than pixels as in most prior works. For this work, we use the standard h.264 codec (Wiegand et al., 2003) and mp4 container (mp4, 2020) to obtain the video bytes. One observation is that these codecs are designed for streaming video. Importantly, this means that when decoding and reconstructing the pixels, the decoder algorithm does it byte-by-byte, or based on short segments of bytes. This means that bytes that are far apart do not really depend on each other to reconstruct the pixels, since they are compressed with streaming in mind and refer to different frames. As a result the standard global self-attention in Transformer models is not needed to understand video bytes. This is a unique characteristic of compressed video bytes, which is not present in text sequences and language modeling. However, to understand a video, i.e., for classification or question answering purposes, the model does need to be able to understand the whole sequence. To address this, we develop techniques which (a) work with extremely long sequences, (b) use sequential modeling to better model streaming video bytes, rather than self-attention as Transformers use and (c) a hierarchical model that is able to go from low-level bytes to semantic understanding. This allows understanding both low-level local features as well as high level details of the full video. Our approach is also generic, we simply input byte sequences without providing any details about the structure, such as i or p frames, and let the model learn with no given priors or biases, an approach that has been important in language modeling. This model is shown in Fig. 1 and described further in Sec. 3. Another issue is that the model tends to overfit and not generalize well when given bytes as input, which we address in Sec. 3.2.

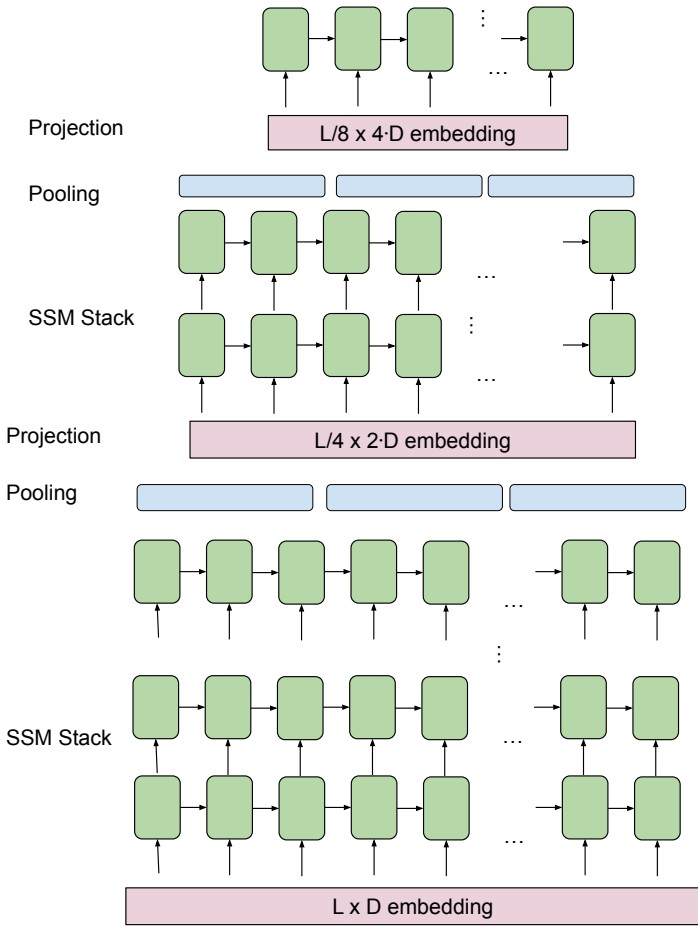

Input Bytes (L): 0000001A6764001FACD9A014016E0000000168EBCCB22C....

Figure 2: The multilevel SSM. The input is processed sequentially by SSM layers. The pooling layers reduce the sequence length; projection layers increase the dimensionality. This process is repeated multiple times, forming the multilevel SSM. This is further parallelized in order to handle long sequences (Sec. 3.1).

On the other hand, learning from raw bytes presents opportunities for learning from much more economical and efficient compressed video formats which are already readily available, as videos are stored in these formats. Additionally, the approach directly transfers to audio+video inputs, as shown in the experiments. We further save compute in the input pipeline, as we do not need to decode the bytes into pixels , though for larger models this is not the bottleneck, as the compute time of the model is the primary bottleneck.

## 3 Proposed Approach - Parallelized Multilevel SSM

To learn from very long sequences, we first design a multilevel SSM module, which has a better handle over the sequence information. Recent SSM models (Gu et al., 2021; Gu & Dao, 2023; Gu et al., 2022b), which scale linearly and are more suitable for long sequences, are still not able to fully address some main challenges, such as long-range recall, recency bias, global sequence understanding, and still have some inefficiencies, e.g., VideoMamba (Li et al., 2024b) trained on only 8 frames and evaluated on up to 64 frames. We choose to build upon an SSM for a few reasons. First the SSM (Gu et al., 2021) does not use self-attention, thus it scales linearly to long sequences. Second, the SSM processes the bytes sequentially as they are input. Since video codecs like mp4s were designed for streaming videos, an SSM is naturally applicable to this input, processing the bytes and updating the state in the order the bytes are input. Specifically, we use the Mamba (Gu &

Dao, 2023) architecture as the base model, given its strong performance among existing SSMs, hardware efficiency, and ability to vary the representation with time (Gu et al., 2021; 2022b; Gu & Dao, 2023; Chen et al., 2024). An overview of the multilevel SSM is shown in Figure 2. Given an input sequence, $S$, that consists of $L$ bytes, we first embed the bytes as $D$-dimensional vectors, which are input to Mamba. We then apply $M$ standard Mamba layers. After this, we pool the bytes which decreases the sequences from length $L$ to length $\frac{L}{L_s}$, e.g, $L_s = 2$ to reduce it be half. We explore different forms of this pooling. We repeat this stack of $M$ Mamba layers, followed by a pooling layer $N$ times, forming the multilevel SSM. Finally, we average pool over the remaining tokens and a fully connected layer for classification tasks. Compared to a prior hierarchical SSM (Bhirangi et al., 2024), the key differences is that our SSM is over the whole input sequence, rather over segments and we stack many levels (4 in our experiments) of the pooling, rather than just two levels. Since SSMs scale linearly with sequence length, there is no benefit to splitting the sequence into segments when running the SSM, and applying the SSM to the whole sequence allows the model to have knowledge of the whole sequence through the SSM state.

For the embedding, as there are 256 bytes, we use a vocabulary size of 256 tokens with an embedding dimension ($D$) of 256. We explore the embedding dimension in the ablations. We note that this embedding dimension increases, more memory is used. Despite the fact that using 256 dimensions has more expressiveness than the original 256 bytes, we found using fewer dimensions greatly reduced performance, and going above 256 did slightly improve performance, but also increases memory usage. To address this, we make a few important steps: 1) make the input to the embedding a uint8 type, rather than an int32 as is standard in LLMs (due to their larger vocabulary size). This saves ∼75% of memory by needing only 8 bits per token; 2) we use 16-bit precision on the embeddings themselves, finding no difference to 32-bit precision, but saving memory. We evaluate attention, averaging and concatenation as pooling methods (see the Appendix for details).

## 3.1 Handling Long Sequences with Parallelism

Due to the very long sequences that video bytes have, we propose a sequence parallelism technique, utilizing multilevel SSM states. We observe that the SSM outputs can be computed for any part of the sequence without having access to the prior state, then propose a simple update applied when given the state. This allows us to parallelize by placing subsequences on different devices and computing the SSM on each device, then applying an update on the outputs, once each device is done computing. This allows cheap parallelism along the sequence axis, which allows us to scale more efficiently to long sequences. We find that sequence parallelism is around 2x faster than model parallelism for this SSM. Namely, we compute and store the hidden states for each device as if the sequence starts with hidden state set to zero. Consider a signal with $L$ tokens $x = [x_1, x_2, ..., x_L]$ and $V$ devices, we divide the signal into $V$ subsequences each of length $K = \frac{L}{V}$ and apply SSM on each device independently, i.e., the initial hidden state is set to zero for every device. The final hidden states $h^{(i)}$ (Eq. 1) and outputs $y^{(i)}$ of the SSM (Eq. 2) applied to each of the $V$ subsequences is as given below. Given $A, B, C$ are the parameters of the SSM and $h$ is the state:

$$
\begin{aligned}
h^{(1)} &= Bx_K + ABx_{K-1} + \cdots + A^{K-1}Bx_1 \\
h^{(2)} &= Bx_{2K} + ABx_{2K-1} + \cdots + A^{K-1}Bx_{K+1} \\
h^{(V)} &= Bx_L + ABx_{K-1} + \cdots + A^{K-1}Bx_{(V-1)K+1}
\end{aligned}
\tag{1}
$$

$$
\begin{aligned}
y^{(1)} &= [CBx_1, CBx_2 + CABx_1, \ldots, CBx_K + CABx_{K-1} \\
&\quad + \cdots + CA^{K-1}Bx_1] \\
y^{(2)} &= [CBx_{K+1}, CBx_{K+2} + CABx_{K+1}, \ldots, CBx_{2K} \\
&\quad + CABx_{2K-1} + \cdots + CA^{K-1}Bx_{K+1}] \\
y^{(V)} &= [CBx_{(V-1)K+1}, \ldots, CBx_{VK} + CABx_{VK-1} \\
&\quad + \cdots + CA^{K-1}Bx_{(V-1)K+1}]
\end{aligned}
\tag{2}
$$

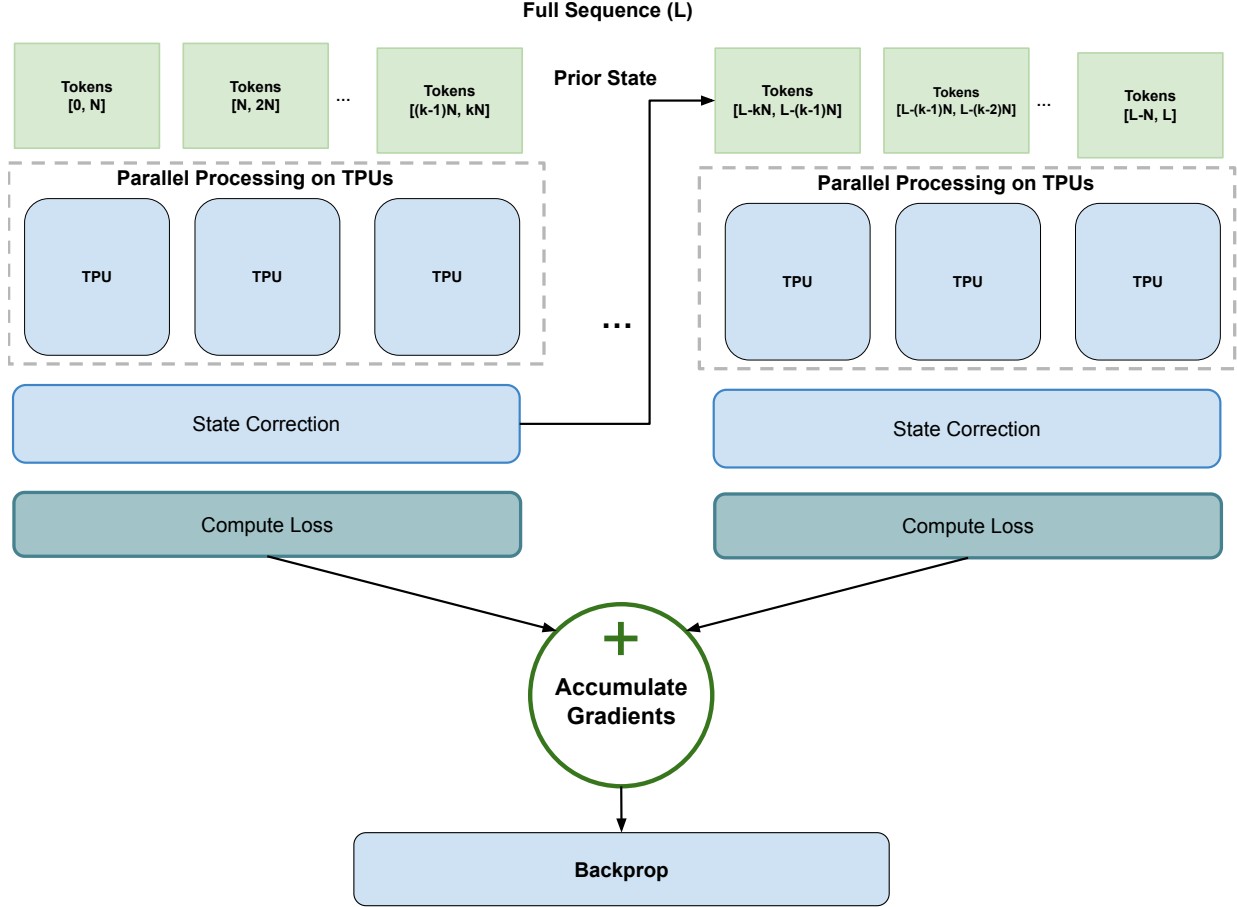

Figure 3: Using sequence parallelism and gradient accumulation to train on sequence lengths of up to 15 million. A key benefit of the SSM is parallel computation of the states, plus a cheap correction calculation (Sec. 3.1), making it easy to scale and suited for streaming video byte data.

The above hidden states are computed in parallel. As observed, they lack the contribution of state from the previous chunks, which are computed on other devices. Thus, the final hidden state from previous sequence shard is added to correct for the missing factor, following the sequence parallelism introduced by Gu & Dao (2024); Smith et al. (2023). We apply this principle to multi-device distribution for very long video byte sequences. The equations for corrected hidden states are given in Eq. 3.

$$
\begin{aligned}
h^{(1)}_{corrected} &= h^{(1)} \\
h^{(2)}_{corrected} &= A^K h^{(1)} + h^{(2)} \\
h^{(V)}_{corrected} &= A^K h^{(V-1)}_{corrected} + h^{(V)}
\end{aligned}
\tag{3}
$$

Here $h_0$ is the hidden state to the entire sequence before splitting into subsequences. The equations for the output after the inclusion of correction factor to the hidden state is as given below (Eq. 4):

$$
\begin{aligned}
y^{(1)}_{corrected}[i] &= y^{(1)}[i] \\
y^{(2)}_{corrected}[i] &= CA^{(i)} h^{(1)}_{corrected} + y^{(2)}[i] \\
y^{(V)}_{corrected}[i] &= CA^{(i)} h^{(V-1)}_{corrected} + y^{(V)}[i]
\end{aligned}
\tag{4}
$$

Here $i \in [1, K]$ is the index of the output within the given subsequence. As shown, this allows us to compute the SSM in parallel across the devices, then correct the state afterwards, with a cheaper correction calculation. We also note here that we are building on Mamba which has the selection mechanism, however this mechanism does not depend on the states, only on the inputs, so it is unaffected by the changes we make here.

**Training on even longer sequences.** Building on that, we further push the limits to train on sequence lengths of up to 15 million using gradient accumulation. Here, we split a very long sequence into subsequences that are as long as we can fit into device memory. We run the first subseqeunce and compute the loss, but rather than taking an optimization step, we save the gradients, and run on the next subsequence using the last state $h$ as the current state on the next subsequence (Fig. 3). We can then repeat this process as many times as needed to reach any length sequence, which enables learning from very long sequences. We note that this gradient accumulation approach for scaling, can theoretically introduce nonlinearities, due to the arbitrary boundaries in splitting the sequence. However, in practice we find that this is not a major limitation and the model is able to learn effectively from these long-range signals, and performs better than training without them (Table 1, 2).

### 3.2 Addressing Overfitting

We find that training data and augmentation is especially important. We observed that the model very easily overfits when trained on encoded videos. However, if we train on individual frames encoded on JPEGs, we did not observe this overfitting. We realized that when training on individual frames we randomly sampled one of the frames, and as Kinetics videos are 10 seconds long, and we had data at 25 fps, we had roughly 250 frames we were sampling from, i.e., we roughly increased the data by 250 times. If instead we trained the model using only a single JPEG byte string per video, e.g., the 100th frame, we saw the same overfitting behavior. In both these cases, when trained on a single encoded video or single encoded frame, the loss would go to 0 very quickly, while the evaluation accuracy would be relatively low, regardless of input format. This indicates the model needs significantly more training data, either in the form of data augmentation and/or pre-training. We find that straightforward data augmentation mitigates these effects.

Our next observation is that if we take two video clips and apply mild data augmentation, e.g., some slight color jittering, as is standard in training video models (e.g., ViViT (Arnab et al., 2021)), and encode them into mp4s, and compute the Levenshtein distance on the byte strings of these clips, we saw that over half the byte string is different. Visually, the two videos are indistinguishable. This suggests that the compressed Bytes-based representations have a large amount of variation even with seemingly small changes to the visual inputs. Because of this, when training video byte based models, we apply a large amount of data augmentation during training. Specifically, we apply random temporal and spatial cropping, color jittering, contrast adjustment, color inversion, posterization, solarization, brightness, sharpness, and cutout augmentations. These augmentations are all done on the RGB space, then encoded into mp4s and used to train the model. We don't apply any augmentation to the bytes themselves (e.g., byte-level dropout, random substrings, etc), and leave explorations of that for future work.

**Self-supervised Pre-training.** We also explore self-supervised pre-training for video byte based models. Byte-based representations enable many different fully self-supervised tasks. First, we can train a model that takes the encoded video bytes as input and produces the RGB pixels of the video as output. However, as video generation is a complex task, itself having many specialized models and methods which are computationally intensive (e.g., video diffusion), and since we don't care about actually generating videos, just about training a video understanding model on Byte-based representations, we simplify the RGB prediction significantly. Here, we take the output of the multilevel SSM model, and use an attention pooling layer to generate $F' \cdot H' \cdot W' \cdot D$ tokens. This is then reshaped to $F' \times H' \times W' \times D$, which gives us the rough size of the video. We then apply a small UNet-based (Ronneberger et al., 2015) model to upsample this to the video tensor $F \times H \times W \times 3$. We then apply a MSE loss between the predicted video and ground truth video RGB pixels. To further reduce compute, we generate only 10 frames per video (i.e., 1 fps for 10 second clips) at a low resolution of $128 \times 128$. While the reconstructions do not look perfect, this provides a good enough learning signal to the model. Second, we explore a pre-training similar to how language models are trained:

Table 3: Kinetics-600 and Kinetics-400 results. We note that our model is significantly cheaper than prior works and trained on much less data. PT stands for Pre-training.

| Model | PT Data | PT Modalities | Params | TFLOPS | K600 | K400 |
|---|---|---|---|---|---|---|
| ViViT-L (Arnab et al., 2021) | JFT-300M | Img | - | - | 82.9 | 83.5 |
| ViViT-H (Arnab et al., 2021) | JFT-300M | Img | - | - | 85.8 | 84.9 |
| MerlotReserve-H (Zellers et al., 2022) | YT-1B | Vid+Audio+Text | 644M | - | 91.1 | - |
| TubeViT-H (Piergiovanni et al., 2023) | ImageNet | Img | - | 17.64 | 91.8 | 90.9 |
| InternVideo2 (Wang et al., 2024b) | Many | Img+Vid+Audio+Text | 6B | - | 91.9 | 92.1 |
| VideoMamba (Li et al., 2024b) | CLIP-400M | Img+Text | 74M | 28.42 | - | 85.0 |
| VideoMAE (Tong et al., 2022) | None | - | 600M | 88.76 | - | 87.4 |
| ST-MAE (Feichtenhofer et al., 2022) | IG-uncurated | Vid | 600M | 25.1 | - | 86.8 |
| BytesSSM-B (ours) | HowTo100M | Vid | 500M | 1.88 | 60.5 | 61.3 |
| BytesSSM-L (ours) | HowTo100M | Vid | 1B | 4.12 | 85.2 | 86.8 |

Table 4: Results on longer video understanding on AcitivityNet-QA (ANet) and CinePile benchmarks.

| Model | ANet | CinePile |
|---|---|---|
| VideoCoca (Yan et al., 2022) | 56.1 | - |
| UMT-L (Li et al., 2023) | 47.9 | - |
| Mirasol-3B (Piergiovanni et al., 2024) | 51.1 | - |
| LLaVA-OV-7B Li et al. (2024a) | 56.6 | 49.3 |
| BytesSSM-L (ours) | 57.1 | 47.5 |

next byte prediction. I.e., the input to the model is a encoded mp4 byte string and the models task is to predict the next byte based on the previous bytes.

## 4 Experiments

### 4.1 Main results

We present the main experimental results on the Kinetics-400 (Carreira & Zisserman, 2017), Kinetics-600 (Carreira et al., 2018), AcitivityNet-QA (Yu et al., 2019), CinePile (Rawal et al., 2024), MLB YouTube (Piergiovanni & Ryoo, 2018), and Kinetics-Sounds (Arandjelovic & Zisserman, 2017) benchmarks, which are popular video or video+audio understanding benchmarks. Please see the Appendix for implementation details.

**Kinetics-400/600.** Table 3 shows the classification performance of the proposed method on the commonly used activity understanding benchmarks, Kinetics-400/Kinetics-600, with 400/600 classes. We note that Kinetics has 10 second videos, and we used 25 fps (far higher than previous works) giving us 250 frames per video, but after encoding, the input is only approximately 250,000 bytes long. We also note that prior works benefit from image-based pre-training which is beneficial, while we here only use video pre-training. Our results show strong performance, even when compared to larger video and video foundational models, despite using far less data and compute. Our model uses only 2-4 TFLOPs, much fewer than others including SSM-based models.

**Long Videos.** Table 4, further shows performance on longer videos which is clearly very challenging for raw bytes formats. The ActivityNet-QA dataset has videos with the average duration between 5 and 10 minutes, whreas CinePile videos average 3 minutes of duration, with some videos as long as 8 minutes. The task is Video QA which requires our model to output free-form text (see the appendix for details on the setup and evaluation).

Table 5: Results on the MLBYouTube benchmarks which is a high-fps dataset for fine-grained video understanding.

| Model | Params | MAP |
|---|---|---|
| MLBYT (Piergiovanni & Ryoo, 2018) | - | 62.6 |
| BytesSSM-B (ours) | 500M | 63.5 |
| BytesSSM-L (ours) | 1B | 64.5 |

Table 6: Kinetics-Sounds (Arandjelovic & Zisserman, 2017) results. We see that including audio in the encoded video helps.

| Model | +Audio | MAP |
|---|---|---|
| MBT (Nagrani et al., 2021) | Yes | 85.0 |
| BytesSSM-L (ours) | No | 81.4 |
| BytesSSM-L (ours) | Yes | 84.4 |

The videos in these tasks reach 15 Million training bytes sequence lengths, which by itself is a particularly long input sequence length, due to long-sequence challenges such as signal attenuation, recency bias and others. We demonstrate it is feasible with our proposed model here.

We note that the current performance of our bytes-only model shows competitive results, but is also outperformed by some SOTA models, e.g. LLaVA-OV-7B (Table 4). Considering its size, e.g. 1B and its raw formats and the challenge of understanding long videos, this performance demonstrates its feasibility in these scenarios, as well.

**High-Fps Videos.** Table 5 further evaluates the performance on the MLB Youtube benchmark (Piergiovanni & Ryoo, 2018), which is a benchmark for distinguishing between fine-grained activities, which requires understanding motion at higher FPS than other datasets, like Kinetics. As seen, the model performs very well, outperforming the state-of-the-art approaches, using encoded bytes as input.

**Audio+Video.** Our model easily extends to Audio+Video from bytes, showing competitive performance (Table 6).

## 4.2 Model efficiency

One advantage of the proposed approach is the compute and time savings, since it operates on a highly compressed format. In Figure 4, we present the results comparing the FLOPs and runtime of the proposed model. As seen, our models are much more lightweight. Furthermore, with other approaches a very limited number of frames can be sampled, e.g., up to 32, and it becomes prohibitively expensive to sample more frames for these models, whereas here the information content of all frames is processed which exceeds the upper bounds over number of frames for other models, e.g., all 250 frames on Kinetics (10 second clips at 25fps) for ours vs 32 or 64 for prior works.

**Parameter Efficiency Discussion** Encoded video bytes are a compressed representation and as such we found it took more parameters and training iterations to match the performance of pixel-based models. Models operating on pixels start with a representation that is already structured for (human) perception. Nearby pixels are spatially related, and basic patterns (edges, textures) are immediately available, and that is used by convolution/patch based models. However, starting from bytes, the model must first implicitly learn the 'language' of the codec. Thus some portion of the model's parameters are dedicated to solving this problem of 'decoding' the byte stream into a useful latent representation, which pixel-based models don't have to do, instead starting with the inductive bias of convolution/patches. In some sense, a Bytes-based model is doing more with each parameter, since it has no inductive bias. We note that the FLOPs, runtime and memory usage of the model is significantly lower than pixel-based ViTs, despite the larger parameter count. Since the relationship between FLOPs, runtime, memory usage, and parameters depends a lot on the

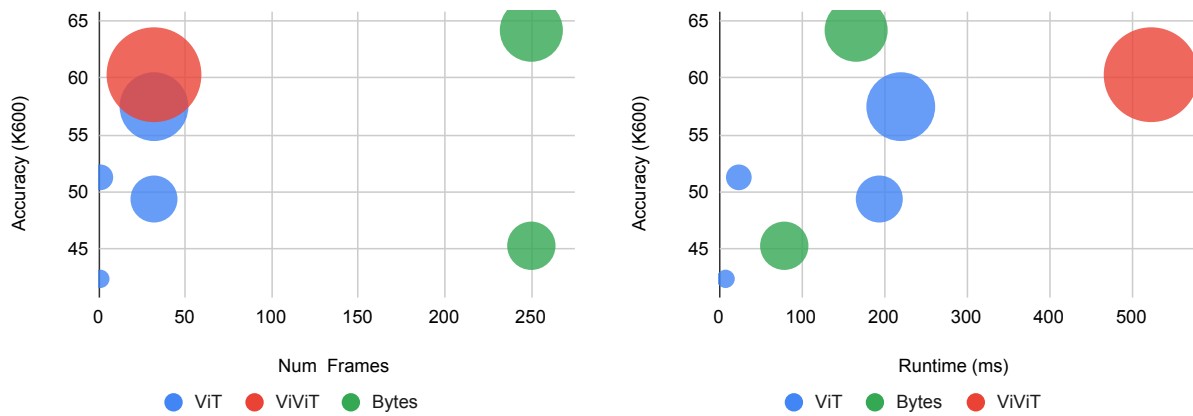

Figure 4: Plot of accuracy vs. runtime and frames, where the size of each model indicates how many FLOPs it uses. This shows the Bytes models scale better to longer sequences, use fewer FLOPs, even as the model scales up, showing the potential of this approach.

Table 7: Scaling ablation on Kinetics using BytesSSM-Tiny. Increasing the embedding dimension of the model leads to overfitting.

| Model | K600 |
|---|---|
| 256 Embedding | 25.4 |
| 512 Embedding | 25.7 |
| 1024 Embedding | 21.2 |

Table 8: Pooling method experiments, using BytesSSM-Tiny.

| Method | TFLOPs | K600 |
|---|---|---|
| Attention Pooling | 1.33 | 23.9 |
| Average Pooling | 0.38 | 20.7 |
| Concatenation Pooling | 0.47 | 25.4 |

network structure, the multilevel SSM on Bytes-based data enables these compute savings even with more parameters.

## 4.3 Ablations

For the ablations, we report the accuracy on Kinetics-600 using the BytesSSM-Tiny model.

**Model ablations**. Table 7 shows how the model scales with size, particularly, with increasing the size of the dimensionality of the feature representation in the model, which also results in larger model. As seen larger feature representation is beneficial, however, we observe overfitting for very large model sizes. Table 8 compares different versions of the pooling. We use concatenation pooling as it performs best for relatively small increase in FLOPs.

Table 9 explores several versions of the proposed model. As seen, using SSM-style models provides benefits for this application due to the sequence lengths and nature of the encoded byte structure, compared to a Transformer (Vaswani et al., 2017) model. The proposed Multilevel SSM provides a further large jump in performance, as it has much higher capacity for increased sequence lengths. We also compare to a Multilevel Transformer, which uses the same pooling methods as the Multilevel SSM, but replaces the Mamba blocks with standard Transformer blocks.

We further compare to a causal convolution based model, as well as, the one proposed in (Horton et al., 2023), which uses many pooling layers and sliding window attention for encoded e.g., JPEG/PNG images. We note that here we are using longer sequences than those tested in that paper (262,144 vs. 150,000). For

Table 9: Comparing different forms of the model. A standard Transformer with full global attention did not fit into memory with the long sequence lengths videos have (BytesSSM-Tiny model).

| Model | K600 |
|---|---|
| Transformer | N/A |
| Transformer with loc. attention with 512 tokens | 15.4 |
| Causal Conv ($\sim$ same params as BytesSSM-Tiny) | 15.8 |
| BF-Ti Horton et al. (2023) | 17.5 |
| Multilevel Transformer | 16.7 |
| Mamba (Baseline SSM) | 18.4 |
| BytesSSM-Tiny (with Multilevel SSM) (ours) | 25.4 |

Table 10: Effects of pre-training tasks, using the BytesSSM-Tiny model, on HowTo100M.

| Method | K600 |
|---|---|
| None | 25.4 |
| RGB prediction | 31.2 |
| Next-byte prediction | 28.6 |

Table 11: Training on JPEG bytes and transferring to mp4 bytes, using BytesSSM-Tiny model.

| Method | K600 |
|---|---|
| No Pre-training | 25.4 |
| JPEG Pre-trained | 27.2 |
| mp4 Pre-trained | 31.2 |

Table 12: Effects of data augmentation. Tiny model.

| | K600 |
|---|---|
| 1x data | 2.4 |
| 10x data | 5.2 |
| 100x data | 18.6 |
| 200x data | 25.4 |

both, we used models matching the same parameter count as BytesSSM-Tiny. In Table 9 (Lines 3, 4), we find that our approach significantly outperforms these variants, as well.

**Pre-training Experiments**. Table 10 explores different methods for pre-training for the proposed models. We compare RGB reconstruction to next-byte prediction, finding that RGB reconstruction is slightly better as a pre-training task, but both are effective.

We also see how transferable Byte-based models are. In Table 11, we compare no pre-training to a model pre-trained for RGB reconstruction vs a model pre-trained on JPEG bytes for classification. Previously, most video works used image pre-trained backbones which were then further trained on video data, and this experiment is similar to that. We see some benefit from pre-training with image JPEG bytes, compared to no pre-training, but it is not as good as video byte based pre-training. This is expected, but also shows these models are learning some generalization knowledge about encoded byte structures, even for very different encodings.

**Data Augmentation Effects**. Table 12 shows the effect of data augmentation when training. We generated a fixed number of samples by applying data augmentation as described above to generate 1 to 200 samples for each video. We find that augmentation is an important component of training and that increasing the data augmentation increases the performance of the model a lot. We further note that small amounts of augmentation are not yet helpful. Augmentation seems to be of positive effect to models of all sizes.

**Preprocessing Cost Comparisons** We further compare the different preprocessing steps and times. In some cases, videos are stored in different formats, so a one-time transcoding operation may be needed. For these comparisons, we used the Kinetics videos. For the bytes input pipeline, we had a one-time transcoding cost of 193ms per video, 10ms to load the mp4 into memory, <1ms to transfer it to the TPU/GPU, for a total of 203ms. For pixel-based pipelines, we had 180ms to decode a mp4 into RGB pixels, 69ms for crop, resize, etc. 4ms to transfer 64 frames to TPU/GPU, a total of 254ms. These are very similar. The transcode, though, only needs to be done once, so for training, our input pipeline is significantly faster.

**Training Efficiency Comparison** As prior works have a large variance between pre-training data, methods, model sizes, etc. we here provide a comparison of a standard ViT-H trained on the HowTo100M RGB pixel data vs. our proposed Bytes-SSM on encoded video bytes. This allows us to directly compare a standard baseline model to ours, keeping the data and model size consistent. We train both models using the same number of TPUs (128) and compare for keeping training iterations the same or keeping the train wall-clock time the same. The results are shown in Table 13. We use ViT-H as it has a similar number of parameters to

| Model | Training Constraint | K600 Acc. |
|---|---|---|
| ViT-H (Pixels) | 1M steps | 54.6 |
| BytesSSM-B | 1M steps | 48.8 |
| ViT-H (Pixels) | 2M steps | 60.3 |
| BytesSSM-B | 2M steps | 60.5 |
| ViT-H (Pixels) | 30 hours | 55.3 |
| BytesSSM-B | 30 hours | 48.5 |
| ViT-H (Pixels) | 61 hours | 60.2 |
| BytesSSM-B | 61 hours | 60.5 |

Table 13: Training efficiency comparison on Kinetics-600. We compare a standard ViT trained on RGB pixels vs. our Bytes-SSM on encoded bytes using 128 TPUs. Using BytesSSM-B.

| Encoding Setting | K600 Acc. |
|---|---|
| Original (200kbps, 384p, 25fps) | 60.5 |
| *Bitrate Changes* | |
| Lower Bitrate (50kbps) | 58.5 (-2.0) |
| Higher Bitrate (500kbps) | 60.4 (-0.1) |
| *Resolution Changes* | |
| Lower Resolution (180p) | 57.2 (-3.3) |
| Higher Resolution (480p) | 61.2 (+0.7) |
| *Frame Rate Changes* | |
| Lower FPS (10 FPS) | 60.2 (-0.3) |
| Higher FPS (30 FPS) | 60.3 (-0.2) |
| Variable FPS (mp4 encoding setting) | 59.2 (-1.3) |

Table 14: Robustness to encoding changes on Kinetics-600. Despite significant changes to the encoding settings, the model remains robust. Using BytesSSM-B.

BytesSSM-B. Here, we split the training steps or time to be 75% pre-training and 25% fine-tuning. For the ViT models, we use an attention pooling layer to combine the tokens across the frames before classification.

**Robustness to Encoding Changes** We explore how robust the trained model is to encoding changes, such as variable frame rate, resolutions, and bitrates. In Table 14, we show the results, finding that while performance slightly drops when making significant changes, it is overall robust to changes to the encoding settings. Interestingly, when increasing the resolution, the performance slightly increases, similar to performance seen when increasing pixel resolution and evaluating pixel-based models.

## 5 Related Works

**Efficient Video Representations and Compression.** Traditional video compression standards, such as H.264/AVC and HEVC, rely heavily on Discrete Cosine Transform (DCT) for spatial redundancy and motion vectors for temporal redundancy to achieve high compression rates Szczerba et al. (2009); mp4 (2020). Motivated by this efficiency, researchers have explored leveraging these compressed components directly to avoid the computational cost of full decoding. In their foundational work CoViAR, Wu et al. (2018) demonstrated the benefit of training directly on compressed video signals, namely P-frame motion vectors and I-frame residuals, achieving lower computational overhead than traditional RGB-based models. However, CoViAR still relies on standard CNN backbones applied to these partially decoded components. More recently, works such as Deep Residual Learning in the JPEG Transform Domain Ehrlich & Davis

(2019) and Learning from the CNN-Based Compressed Domain Wang et al. (2022) have pushed this further by designing networks that operate mathematically within the compressed domain (e.g., modifying convolutions to accept DCT coefficients), thereby bypassing the decoding step entirely. While promising, these methods often require specialized architectures to handle the non-spatial nature of frequency domain inputs.

**Neural Codecs and Byte-Level Learning.** Parallel to heuristic compression, neural video codecs have emerged as a data-driven alternative. Torfason et al. (2018) proposed learning VQ-VAE models to compress video data, allowing downstream models to operate in a learned latent space. While effective for tasks like action recognition on Kinetics Carreira et al. (2018), this approach shifts the computational burden to the neural codec itself. (Wiles et al., 2023) proposed a compressed vision pipeline for VQ-VAE-based neural codec, learning to represent videos as compressed spatial tensors. Additionally, a learnable augmentation network enables transformations such as cropping and flipping to be learned by the latent codes. At the extreme end of this spectrum, Horton et al. (2023) explored learning from raw image bytes. However, their approach is currently limited by sequence length constraints, effectively restricting it to small images or short clips.

**Efficient Sequence Modeling with State Space Models.** Addressing the computational inefficiency of Transformers, which scale quadratically $\mathcal{O}(L^2)$ with sequence length $L$, State Space Models (SSMs) have gained prominence for their ability to model long-range dependencies with linear complexity $\mathcal{O}(L)$. The Structured State Space model (S4) Gu et al. (2022a) successfully modeled extremely long sequences by constraining the state matrix with HiPPO theory. Building on this, Mamba Gu & Dao (2023) introduced a selective scanning mechanism that allows the model to filter irrelevant information dynamically. In the video domain, VideoMamba Li et al. (2024b) recently demonstrated that purely SSM-based architectures can rival Transformer performance on long video understanding tasks while maintaining linear scaling. This architectural efficiency is critical for processing long temporal contexts that are computationally prohibitive for standard Vision Transformers.

## 6  Conclusions

We propose a novel approach to understand videos from encoded and compressed byte representations. This has the advantage of saving memory and compute, compared to working on pixels, and better scales to longer sequences, reaching 15 Million. We show strong performance on this new and challenging task and demonstrate there is much potential in learning from raw video bytes.

## Impact Statement

This paper presents work whose goal is to advance the field of Machine Learning by presenting an alternative view of learning for video understanding models and exploring if it is possible to learn from raw bytes. As such this research does not introduce any additional potential societal consequences beyond those that are inherent in any video understanding technology.

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

## 7 Appendix

### 7.1 Pooling Methods

Given the input embedding sequence, $L \times D$, we input this to the SSM. Specifically, this model is the Mamba SSM architecture. We do not make any changes to the layers or operations themselves. To create the multilevel SSM, we instead add pooling or merging layers within the SSM. We explore several approaches to this: (1) using attention-based pooling layers, (2) average pooling, and (3) concatenation pooling.

**Attention Pooling.** Here, we pass a sub-sequence of length $L_s$ into the attention layer and output 1 token, thus pooling $L_s$ tokens. For example, given a sequence length of $S$, and $L_s = 4$, we reduce the sequence by 4 by creating a $\frac{S}{L_s} \times L_s \times D$ tensor, then applying attention pooling e.g., (Touvron et al., 2021). Specifically, we use a query vector with 1 latent embedding, when when applied to the input key/value tensor which has size $L_s \times D$, results in a $1 \times D$ output. When applied along the whole sequence, this results in a $\frac{S}{L_s} \times D$ tensor. This can then be passed to the next SSM block.

**Average pooling**. This is similar to the above approach, expect we apply average pooling over the $L_s$ dimension, resulting in the same $\frac{S}{L_s} \times D$ tensor, but without the attention operation, only averaging.

**Concatenation pooling**. Here, we create a tensor of shape $\frac{S}{L_s} \times D \cdot L_s$, by re-arranging the tensor to group multiple tokens into 1 by combining along the embedding axis. I.e., we reshape from $S \times D$ to $\frac{S}{L_s} \times D \cdot L_s$.

Finally, after any of the pooling layers, we apply a fully-connected layer to project the resulting tensor to the final dimension $D_{out}$, which can either be the same as $D$ or larger. We found $L_s = 4$ and $D_{out} = 2 \cdot D$ worked well in our experiments. This reduces the memory used by the sequence by a factor of 2 each time a pooling layer is applied.

### 7.2 Implementation Details

Table 15: Model configs used in the paper.

| Model | Layers | $D$ | $L_s$ | $N$ | $M$ | Params | TFLOPs |
|---|---|---|---|---|---|---|---|
| Bytes-Tiny | 12 | 256 | 4 | 4 | 3 | 103M | 0.47 |
| Bytes-B | 33 | 256 | 4 | 4 | 8 | 500M | 1.88 |
| Bytes-L | 45 | 256 | 4 | 4 | 11 | 1B | 4.12 |

We encode the videos as H.264 and in mp4 containers with a image size of 384 and a bitrate of 200kbps. This roughly matches resolution standard video models use. The model architectures used, Bytes-Tiny, Bytes-Base (Bytes-B), Bytes-Large (Bytes-L), are described in Table 15. Unless otherwise noted, we use only the video stream and do not include audio in the encoded video. We find the model is sensitive to learning rates, both different tasks (e.g., pre-training vs. classification finetuning) and model scales need different learning rates. We use $9e^{-5}$ as the pre-training learning rate for the Bytes-Tiny model, $7e^{-5}$ for the Bytes-Base model and $5e^{-5}$ for Bytes-Large. For fine-tuning, we use $5e^{-5}$, $3e^{-5}$, and $1e^{-5}$ for the tiny, base and large models, respectively. We pre-train with a batch size of 64, a sequence length of $2^{1}8$ (262144) and for 2,000,000 steps. We fine-tune with the same settings, but for 1,000,000 steps. We note that with sufficient data augmentation, we do not observe overfitting behaviors even with 200 epochs of training on Kinetics-600. We use the Adam optimizer (Kingma, 2014), which is also important, with default settings. We use 512 TPU v5p to train the model. The Tiny model runs at approximately 20 steps per second, Base runs about 9 steps/sec and large about 4 steps/sec. Thus to train the model it takes about 27 hours, 61, and 127 hours to pre-train the models respectively. And about 14, 30, 63 hours for fine-tuning.

Table 16: Experiment showing performance of different video codecs and containers, using Bytes-Tiny. We see there is a small difference between the settings, but in general, they all perform very similarly.

| Codec | Container | K600 |
|-------|-----------|------|
| h.264 | mp4 | 25.4 |
| h.265 | mp4 | 25.1 |
| h.264 | mov | 25.8 |
| h.265 | mov | 24.7 |
| VP9 | mp4 | 24.3 |
| VP9 | WebM | 24.9 |

Table 17: Multilevel SSM applied to the whole sequence vs. Chunked sequence (as in (Bhirangi et al., 2024). We note that there is no noticeable difference in FLOPs or compute time between these two approaches.

| Model | K600 |
|-------|------|
| Ours | 25.4 |
| Chunked (Bhirangi et al., 2024) | 25.1 |

In Table 15 we give the details for each model configuration used in the paper.

**Long Video Training** To reduce compute costs, we train the model in stages. First, we do the pre-training as above on short video segments. This provides us with a good base model that understands video bytes. Next, we do two stage of long video training, using the method described in Section 3.1. We train on the VideoMarathon (Lin et al., 2025) data in two stages, first with sequence lengths of 2 million then 15 million bytes, roughly 1.3 minutes and 10 minutes long. We emphasize here that we are training on the full video, without any subsampling, unlike prior works.

## 7.3  Additional Experiments

In Table 16, we compare the model using different codecs and containers. While these results show the performance is pretty similar across codecs, it is possible some are easier for the model to learn than others.

In Table 17, we compare our proposed multilevel SSM to the chunked based one in (Bhirangi et al., 2024), the main difference being we apply the SSM to the entire sequence, while the other chunks the sequence and then applies the SSM independently to each chunk. Due to our efficient implementation of sequence parallelism, there is no meaningful difference in compute costs or runtime between the approaches, and for video byte inputs, applying the SSM to the whole sequence is better.

# 8  ActivityNet-QA and Cinepile Long Video Experiments

For the experiments in table 6, we needed to add a language model to handle the question answering task. To do this, we used the Gemma model Team et al. (2025a). Specifically, we then took the final representations from the SSM model as the video representation, and added that to the embedded text representations and then trained Gemma to generate the answers. The Gemma model is already pre-trained from the public version used. Subtitles are not used as part of the evaluation.

For CinePile, since it is a multiple choice dataset, we evelute using standard accuracy, if the predicted answer (e.g., a, b, c, or d) matches the ground truth.

For ActivityNet, since it is open-ended questions and the ground truth is in the form of text, we output a free-form text as an answer. We use string equality to compare the answers, that is, the predicted output, after normalization, is directly compared for an exact match to the ground truth. If the output matches exactly the ground truth output, the example is counted as correct, and as incorrect otherwise. In terms of output normalization, we apply the standard stripping of whitespaces, lowercase normalization, articles and

punctuation removal. While this is a more challenging evaluation setting, we prefer it as it is more natural, and is in line with evaluation of prior methods.

CinePile videos average 3 minutes of duration, with some as long as 8 minutes. It has both a training and evaluation set, so we finetune the 15M token model on this data for 1 epoch. We train the entire model with a learning rate of 0.00001 on this question answering task.

ActivityNet-QA has videos with the average duration between 5 and 10 minutes. It also has a training and evaluation set, and we finetune for 1 epoch as well with a learning rate of 0.00001.

## 9   Discussions on Additional Pre-training

There are many other pre-training tasks could be explored, such as codec translation, e.g., mp4 (H.264) as input and generate a VP9 encoded video as output, using a standard per-token cross-entropy loss, or predicting features from a known visual encoder (e.g., CLIP (Radford et al., 2021) features) rather than directly predicting pixels. Similarly contrastive losses across different codecs could be used. Weakly supervised tasks such as predicted ASR transcripts from video byte inputs could be explored. We leave these explorations as future work.

## 10   Discussion on Augmentation and Efficiency Trade-offs

We here discuss the practical costs of the heavy data augmentation (up to 200x) required for byte-level learning. We provide a breakdown of these costs here.

**Storage Overhead**: A typical 10-second Kinetics clip at 200kbps is approximately 250,000 bytes in encoded mp4 format. With 200x augmentation, the total storage per original clip increases to 50 MB. For the Kinetics-600 dataset ( 400,000 clips), this corresponds to roughly 20 TB of storage. While significant, this is still an order of magnitude smaller than the storage required for the equivalent RGB pixel representations (e.g., PNGs or JPEGs), which would occupy approximately 166 TB for the 400,000 clips.

**Computational Cost**: On-the-fly augmentation involves a decoding, transformation, and re-encoding step. Our pipeline performs this in approximately 203ms per sample. This is notably comparable to or faster than the standard RGB preprocessing pipeline (254ms), which includes decoding, cropping, and resizing. Thus, while byte-level models require more training samples due to lack of spatial priors, the per-sample processing cost remains competitive.

**Fundamental Limitation vs. Structural Priors:** We observe that byte-level representations are extremely sensitive to visual changes; a mild RGB perturbation can alter over half the byte string due to entropy coding. This suggests that the augmentation is not merely a data-scaling technique but a necessary 'behavioral prior' that teaches the model to ignore codec artifacts and focus on semantic content. While adding explicit codec-aware structural priors (e.g., I-frame markers) might reduce this dependency, our results demonstrate that a purely format-agnostic model can succeed given sufficient augmentation.

