# OpenReview forum: "Learning to Understand Videos From Encoded Bytes"
_TMLR — Accepted by TMLR_

### Review · Reviewer_b55m · 2026-04-19

**Summary Of Contributions:**

This paper introduces BytesSSM, the first approach to video understanding that operates directly on raw encoded byte streams (H 264/mp4 format) rather than decoded RGB pixels. The motivation is grounded in a practical observation.. compressed video is roughly 99% smaller than its decoded equivalent, and current systems either discard this efficiency advantage by decoding into pixels or heavily subsample frames to fit memory, both of which are costly for long or high frame rate video. The authors argue that SSMs are better suited to this input than Transformers because video codecs are designed for sequential streaming, and accordingly build a multilevel Mamba based model with progressive pooling to move from raw bytes to semantic representations. To train on sequences reaching 15 million tokens, they develop a sequence parallelism scheme that distributes subsequences across TPU devices and corrects each device's SSM outputs using the final hidden state propagated from the preceding device. The paper further identifies and characterizes a severe overfitting problem specific to byte level representations, attributing it to the entropy-coded byte string's extreme sensitivity to minor RGB perturbations, and proposes heavy data augmentation and selfsupervised pretraining as mitigations. Experiments demonstrate competitive performance against pixel-based models.

Key strengths are the genuine novelty of the problem formulation, the technically sound sequence parallelism derivation, the clean ablation establishing SSMs' advantage over Transformers and causal convolutions for this specific input type, and the practically important characterization of the data augmentation problem. Key weaknesses are a factual citation error in the related work that misidentifies a prior method, comparison tables that conflate training regime differences with representational advantages, and insufficient analysis of the augmentation dependency as a fundamental limitation of the byte-based approach.

**Audience:**

Yes

**Audience Explanation:**

Video understanding is a central problem in machine learning, and virtually all video in practical deployment is stored in compressed formats. The question of whether models can operate directly on compressed video without decoding is both practically important and theoretically interesting. ByteFormer (Horton et al., 2023) demonstrated this for static images.. extending it to video codecs introduces genuinely new challenges around sequence length, inter-frame temporal structure and training stability, all of which this paper addresses.

The data augmentation finding that mild RGB perturbations alter more than half the mp4 byte string due to entropy coding, causing catastrophic overfitting unless addressed explicitly is a counterintuitive and practically important result that the video understanding community has not previously documented. The sequence parallelism scheme enabling training on 15 million token sequences is of independent interest beyond this application. A substantial portion of TMLR's readership works on efficient sequence modeling, long context learning, or video understanding, and will find this paper's empirical findings and failure mode characterization valuable regardless of whether the byte based approach eventually surpasses pixel-based approaches.

**Broader Impact Concerns:**

The current impact statement ("none which we feel must be specifically highlighted here") does not engage seriously with the paper's implications. The approach's central advantage of processing all video frames rather than subsampling has non trivial compute and energy implications at inference scale that practitioners need to understand. The paper should expand this section to at minimum note that full density processing at scale introduces energy costs commensurate with the frame rate density gains. No other ethical concerns are raised by this work.

**Claims And Evidence:**

Yes

**Claims Explanation:**

The paper's primary claim that video understanding from raw compressed byte streams is feasible for the first time is well supported. Table 9 directly ablates the architectural space and provides principled evidence that SSMs outperform Transformers, localized-attention variants, causal convolutions, and the prior byte-based image model (Horton et al., 2023) on this input type. The sequence length scaling experiments in Tables 1 and 2 concretely demonstrate that longer byte sequences improve accuracy, validating the core motivation. The audio + video extension of Table 6 and the codec robustness study in Table 14 provide useful secondary support. The data augmentation ablation at Table 12 honestly reports near-chance performance without heavy augmentation rather than obscuring this, which is a mark of responsible empirical practice.

However, two specific claims are inadequately supported and require correction before acceptance.

First, the related work contains a factual attribution error that undermines the prior art framing. The paper states: "For instance, CoViAR Wiles et al. (2023) demonstrated the benefit of training directly on P-frame motion vectors and I-frame residuals." CoViAR is the work of Wu et al. (CVPR 2018), not Wiles et al. The paper by Wiles et al. (ACCV 2022, Springer proceedings 2023) is titled "Compressed Vision for Efficient Video Understanding" and proposes a neural compression framework.. it does not train on traditional codec components such as P-frame motion vectors or residuals. The description of Wiles et al.'s contribution is therefore factually wrong on both the authorship and the technical content, and the same error propagates into the Table 3 header where a baseline is mislabeled as "CoViAR." The foundational Wu et al. (2018) paper is not cited at all! A notable omission for a paper whose central claim is novelty over compressed-domain video understanding. This must be corrected.

Second, the comparative performance claims in Table 3 are weakened by an uncontrolled confound. BytesSSM-L (1B parameters, pretrained on HowTo100M video) is placed alongside ViViT-L (JFT-300M image pretraining), VideoMAE (600M parameters, no pretraining), and VideoMamba (CLIP-400M image+text pretraining). None of these baselines matches BytesSSM-L's pretraining data regime. The competitive K400/K600 numbers are therefore suggestive but cannot cleanly support the claim that byte-based representations are advantageous, since the performance difference could plausibly be attributed to the larger pretraining corpus rather than the representational format itself. A matched-regime comparison... a pixel-based SSM baseline trained on HowTo100M under identical conditions would resolve this ambiguity.

**Requested Changes:**

**Critical**

1 - Correct the CoViAR citation error throughout. The related work section and Table 3 header both misattribute CoViAR to Wiles et al. (2023). CoViAR is Wu et al. (CVPR 2018); Wiles et al. (ACCV 2022) is a distinct neural compression paper. The description of Wiles et al.'s contribution is also factually incorrect.. their paper does not train on P-frame motion vectors or residuals. The authors should fix both the attribution and the description, and either cite Wu et al. (2018) in the appropriate position or explicitly justify its exclusion from comparison. This is a matter of factual accuracy.

2 - Provide a matched regime control in the Kinetics comparison. To support the claim that byte representations yield competitive performance, the paper needs at least one strong pixel-based baseline trained on HowTo100M at the same parameter scale and fine-tuned under the same conditions as BytesSSM-L. Without this control, the favorable K400/K600 numbers cannot be attributed to the byte representation rather than to the data and compute advantages of the pretraining setup.

3 - Position the sequence parallelism contribution accurately relative to existing SSM literature. The correction formula in Equations 3 4 is the standard parallel associative scan applied to linear recurrences, which underpins Mamba2's SSD framework (Gu & Dao, 2024) and the parallel prefix scan in S5 (Smith et al., 2023). Neither of these is cited. The contribution of Section 3.1 is the application of this established principle to multi-device distribution for extremely long video byte sequences, which is valuable.. but the framing should accurately reflect that the mathematical structure is prior work. Gu & Dao (2024) and Smith et al. (2023) should be cited and the contribution positioned as an engineering adaptation.

**Strengthening**

1 - Deepen the augmentation dependency analysis. Table 12 is one of the paper's most significant empirical results but is underanalyzed. The authors should address whether this extreme augmentation requirement decreases with larger-scale pretraining data, whether it differs for BytesSSM-L versus Tiny, and whether providing explicit codec structure signals (e.g. I/P frame boundary markers as auxiliary inputs) reduces it. Understanding whether this limitation is fundamental to the compressed domain or addressable with structural priors has significant implications for the approach's practical viability.

2 - Reframe the long video QA results honestly. On CinePile, BytesSSM-L (47.5%) is outperformed by LLaVA-OV-7B (49.3%), and the ActivityNet QA margin is 0.5 points under architecturally dissimilar pipelines (BytesSSM + Gemma vs. end-to-end VLMs trained on diverse multimodal corpora). These results are exploratory rather than competitive, and the paper should frame them accordingly rather than presenting them as evidence of performance parity.

3 - Discuss the codec generalization finding in Table 14. The paper argues that the model must implicitly learn codec structure, yet performance is nearly identical across H.264, H.265, VP9, and WebM.. formats with fundamentally different entropy coding and block prediction schemes. If codec structure learning were a major component of the task, H.265 should be harder for a model pretrained on H.264. The authors should discuss what this robustness implies about the nature of the learned representations.

4 - Revise the broader impact statement and a small number of overclaims. The claim of "essentially unlimited tokens in inference" should be qualified with practical throughput or memory bounds to avoid overpromising. The impact statement should briefly acknowledge the energy implications of full frame processing at scale, which the paper otherwise presents exclusively as an advantage.

---

> ### Author Response · Authors · 2026-05-22
> **Response**
>
> Thank you for your review!
>
> **CoViAR citation:**
> We apologize for this error. Indeed the reference was incorrect due to an identifier mixup which displayed Wiles instead of Wu. We meant to cite Wu et al for CoViAR.  We have corrected this in the new manuscript, cited Wu et al appropriately, expanded on their prior work and  fixed the description for Wiles et al.
>
> **RGB control experiment**
> Thank you for the suggestion, we have added an experiment comparing the proposed approach to a pixel-based ViT baseline using the same training data and method for both pixels and bytes. We have added Table 13 to the paper to show this.
>
> **Position the sequence parallelism**
> Thank you for this comment. We have updated the positioning as recommended, i.e. refined the claims and added the citations on Sequence Parallelism and more details.
>
> **Augmentation experiments analysis**
> Thank you for this comment. We have added further discussion to the paper. Indeed we find that augmentation is important and that this affects all models, with the larger ones to larger extents.
>
> **Long-Video QA Cinepile results**
> This is a fair point that better performances are possible with other models which are trained differently, or are larger models e.g. LLaVA-OV-7B with 7B params. We agree and have revised the discussion around this and agree with the Reviewer. We include this experiment since it is notable that the approach is able to handle longer videos during training (from raw byte inputs only), which has been challenging even for standard models which are readily pre-trainable and with very recent progress.
>
>
> **Codec generalization discussion**
> We want to clarify the experiment in Table 14. The models in Table 14 are all trained from random initialization. With this table, we are showing that the model is not specifically designed or tailored to a specific codec, but when given sufficient training data, it can work on any codec. In Table 11, we show that training on different formats, such as JPEGs does slightly help when then further fine-tuning on mp4s.
>
> **Essentially unlimited claims**
> We agree and have revised these statements and no longer mention this in the paper. Thank you for the suggestion.
>
> **Broader impact statement:**
> Thank you for the comment about broader impact. We have further revised it per your recommendation and also per R 43nK’s note.

---

### Review · Reviewer_43nK · 2026-04-30

**Summary Of Contributions:**

This paper proposes learning video understanding models directly from encoded video bytes rather than decoding videos into RGB frames. The main contribution is a parallelized multilevel SSM architecture, instantiated with Mamba-style sequence modeling, designed to handle extremely long byte sequences. The model uses hierarchical pooling/projection to reduce sequence length across levels, and the paper proposes a sequence-parallel training strategy with state correction and gradient accumulation to scale training to byte sequences of up to 15M tokens and inference to even longer sequences. The paper also studies the importance of data augmentation and self-supervised pretraining for byte-based video modeling.

The key strengths are:
1. The paper explores a novel and interesting direction, video understanding directly from compressed byte streams.
2. The proposed architecture is well-motivated by the sequential/streaming nature of video codecs.
3. The paper provides a broad empirical evaluation across action recognition, long-video QA, high-FPS fine-grained recognition, and audio-video understanding.
4. The ablations on model design, pooling, pretraining, augmentation, codecs, and sequence length help support the feasibility of the approach.
5. The efficiency motivation is compelling, especially for long-video understanding where pixel-based models are forced to heavily subsample frames.

The main weaknesses are:
1. Some claims about compute and efficiency are not fully supported by a complete wall-clock/memory comparison against strong pixel-based baselines under matched hardware and training regimes.
2. The method requires substantial model scale, large-scale pretraining, heavy augmentation, and 512 TPU v5p training, which weakens the practical efficiency narrative.
3. The comparison to existing video models is not always apple-to-apple because prior methods differ substantially in pretraining data, input modality, model size, and evaluation protocol.
4. The paper would benefit from deeper analysis of what the byte model actually learns, whether it is implicitly decoding visual structure, exploiting codec artifacts, or learning task-specific shortcuts.
5. The QA experiments are less clearly described than the classification experiments, especially the integration with Gemma and the exact training/evaluation setup.

**Audience:**

Yes

**Audience Explanation:**

The findings are highly relevant to researchers in computer vision, efficient machine learning, and sequence modeling.

**Broader Impact Concerns:**

The authors have included a brief Impact Statement. The research is foundational and does not introduce specific ethical concerns beyond those inherent in any video understanding technology.

**Claims And Evidence:**

Yes

**Claims Explanation:**

The authors provide strong evidence for scaling up to 15 million tokens and demonstrate that their Multilevel SSM outperforms standard Transformers for video bytes.

However, claims regarding efficiency and "unlimited" sequence length require caution:

- Inconsistent Baselines: Comparisons regarding memory and compute savings are weakened by differences in model size, pretraining data, and sampling methods.
- Efficiency Uncertainty: It remains unclear if the pipeline is practically cheaper than optimized baselines under identical conditions.
- Practical Bottlenecks: While theoretically infinite, inference is likely limited by numerical stability, state degradation, and latency.

Overall, the core feasibility is proven, but efficiency and scalability claims need more rigorous empirical support.

**Requested Changes:**

1. Please provide a more controlled comparison of end-to-end inference cost, peak memory, throughput, and wall-clock runtime against representative pixel-based video baselines under matched hardware and comparable input duration. It would also be useful to separate saved resources from avoiding RGB decoding, compressed input size, and the SSM architecture.
2. While Table 14 touches on different codecs, a more detailed discussion or experiment on how the model handles different bitrates or resolutions during training versus inference would be critical, as these significantly change byte patterns.
3. Provide a more concrete definition of the limitations of theoretically unlimited inference.
4. Many compared methods differ substantially in pretraining data, modalities, model size, and evaluation settings. Please make these differences explicit in the main tables and discussion. A controlled comparison to a similarly sized SSM or Transformer model trained on decoded frame features under the same data budget would greatly strengthen the paper.
5. The integration with Gemma is important but missing details. Please clarify whether Gemma is frozen or fully fine-tuned, how video representations are inserted into the language model, how many visual tokens are used, whether the language model has access to subtitles or only video bytes, and whether the comparison to LLaVA-OV and other baselines follows the same evaluation protocol.
6. The paper would be stronger with qualitative or diagnostic analysis showing whether the model learns visual structure, motion information, codec-specific shortcuts, or dataset artifacts. For example, the authors could test robustness to re-encoding, bitrate changes, resolution changes, frame-rate changes, container changes, or adversarial codec perturbations.
7. Please discuss the tradeoff between heavy data augmentation, required computational resources, and performance more explicitly and report the computational/storage cost of generating 100x or 200x augmented byte streams.

---

> ### Author Response · Authors · 2026-05-22
> **Response**
>
> Thank you for your review!
>
> **Efficiency & Compute:**
> Thank you for this question. We have added some additional experiments to the paper to show the efficiency of the model for our model vs. a standard pixel-based ViT model. We emphasize that despite our model using more compute during training, the savings during inference are significant, and we note that for current LLMs, the inference cost far exceeds the training cost, so inference savings are extremely important to consider. We also note that this is the first work exploring video-byte based learning, and future work could discover training cost improvements, similar to efficiency improvements found for pixel based approaches.
>
> **Different codec formats**
> Per your suggestion, we have added more details for the experiments on different codecs (Table 14) . We note that one strength of our approach is learning in a manner agnostic to the format, which is more general and allows seamless application to new formats. In other experiments in the paper we do see positive transfer across formats which is very encouraging. Thank you for the suggestion.
>
> **‘Unlimited’ tokens at inference**
> We agree with this and have updated the manuscript with regards to ‘unlimited’ tokens at inference and it no longer makes this claim. We acknowledge that as sequences extend toward infinity, the model must still overcome the inherent difficulty of long-sequence length learning problems. Thank you for the suggestion.
>
> **Comparison to other models.**
> We agree with the Reviewer that comparison with other SOTA models is not very easy particularly with the advancements of Large Language and Multi-Modal models due to various model scales, varying levels and stages of pre-training,  different exposure to training data, and pre-training schedules and mechanisms. Our intent is to provide experimentation at a reasonable scale and with publicly accessible training datasets. We have provided further details in the paper where possible. However, even existing works struggle to perform strict comparisons, since the settings of them vary, i.e. VideoMamba, InternVideo, ViViT, Merlot, etc. all use different models, pre-training tasks and data, and model scales, and so unfortunately it is not possible to directly compare to existing models. We provided a comparison in Table 9 using various models on video byte data, and we have added a comparison as Table 13 to the revised paper to compare to matching settings for a standard pixel-based ViT model.
>
> **Gemma details**
> We have provided more details on the Gemma integration in the manuscript. The Gemma model is fine-tuned using the new byte representations; the Gemma model is already pre-trained and publicly available. We use all the tokens from the Bytes-SSM as the visual tokens input to Gemma. Since our BytesSSM has pooling within the model, this is only 4096 tokens. Our evaluation is video-based only - subtitles are not used as part of the evaluation in our case. With regards to the evaluation protocol, we follow the standard protocols established in the literature. Thank you for these suggestions.
>
> **Diagnostic analysis of what the model learns:**
> This is a great suggestion, thank you. We have added experiments to the paper (Table 14) taking the trained model and evaluating it on Kinetics-600 using different bitrates, resolutions and frame rates.
> Data augmentation tradeoffs
> Since we do not use any apriori knowledge of the encoding structure of the videos, which is a much harder task than providing codec-specific information, we demonstrate that augmentations are very valuable in this regime and wanted to report its importance to the feasibility of such an agnostic raw-bytes approach. We acknowledge that this will require more pre-processing or storage, not unlike using additional training data. Nevertheless, the approach is much more light-weight at inference as discussed. We further note that prior training of other formats is another beneficial way of pre-training. These ideas are similar to data augmentation used in training of standard pixel based models (e.g., color jittering, random cropping, etc.) however, it is a bit more expensive to either compute and re-encode the augmentations on-the-fly or uses more storage space to pre-compute them. However, this could be avoided by simply using larger pre-training datasets.
>
> **Impact statement**
> We have also refined the broader impact statement, thank you for your suggestion.

---

### Review · Reviewer_MRVT · 2026-05-01

**Summary Of Contributions:**

The paper studies video understanding directly from encoded video bytes instead of decoded RGB frames, which provides an interesting idea for the community. Compressed video is dramatically smaller than pixel-space video, so learning directly from bytes could make long-context video understanding much more efficient. The paper proposed a multilevel SSM architecture built on Mamba and a sequence-parallel state correction scheme that allows training on very long byte sequences. The method is evaluated on a diverse set of benchmarks. It provides a credible case that nontrivial video understanding from encoded bytes is feasible at scale.

**Additional Comments:**

I think the paper is interesting and can provide insights for the community. I appreciate that it takes on a genuinely difficult problem rather than making a marginal improvement within a standard video pipeline, and the results are also promising. With the aforementioned questions solved, I lean towards accept.

**Audience:**

Yes

**Audience Explanation:**

I believe this paper addresses a genuinely interesting research direction that should be of interest to part of the TMLR audience, especially readers working on efficient video modeling, compressed-domain learning, and long-context sequence modeling. Even independent of acceptance, I think many researchers would want to know that direct video understanding from compressed byte streams appears possible at this scale.

**Broader Impact Concerns:**

No.

**Claims And Evidence:**

No

**Claims Explanation:**

If forced into a binary choice, I would lean No for the current version, although I think the evidence is promising and the paper is directionally strong. The paper does support two important claims well. It shows that learning directly from encoded video bytes is feasible, and demonstrates that the proposed approach can operate on substantially longer sequences than standard video pipelines.

However, I do not think the broader claims around efficiency are fully established yet. The paper reports substantially lower per-forward-pass FLOPs (2–4 TFLOPs vs. 17–88 for baselines) and shorter preprocessing times, but the overall training setup remains very large scale, using 512 TPU v5p and millions of optimization steps. As a result, the current evidence is not sufficient to conclude that byte-level learning is truly more efficient in practice when accounting for the full training budget, as opposed to being cheaper per forward pass. I would have found the efficiency argument much stronger with matched end-to-end comparisons against RGB baselines trained under the same data, hardware, and wall-clock budget.

Similarly, while the results are encouraging, the competitiveness picture is mixed. On the positive side, the method achieves the highest score among compared models on ActivityNet-QA (57.1, vs. 56.6 for LLaVA-OV-7B) and outperforms the prior state-of-the-art on MLBYouTube (64.5 vs. 62.6). On Kinetics-400, it is competitive, matching or outperforming several established models (e.g., ViViT-H at 84.9, VideoMamba at 85.0). However, on Kinetics-600 it remains below most strong baselines, and on CinePile it does not outperform the strongest compared model. I therefore think the paper convincingly establishes feasibility and demonstrates genuine competitiveness on several benchmarks, but the efficiency advantage and broad parity across all settings are not yet fully established.

**Requested Changes:**

1. Strengthen the efficiency analysis by providing a direct end-to-end training budget comparison with pixel-based baselines under matched conditions. While the paper already reports TFLOPs, runtime plots, and preprocessing costs, a wall-clock and total training budget comparison against a matched baseline would make the efficiency argument substantially more convincing.

2. Clarify the long-video QA evaluation protocol. Explain answer normalization, why string equality is appropriate for ActivityNet-QA, and whether the reported comparisons in the long-video table are strictly comparable.

3. Revise the wording of the strongest efficiency claims. In particular, phrases suggesting essentially unlimited inference length should be better qualified. While the paper demonstrates extrapolation from 15M to 20M tokens (Table 2) and the gradient accumulation mechanism theoretically supports arbitrary lengths, more extensive evidence at even longer sequences would strengthen this claim.

---

> ### Author Response · Authors · 2026-05-22
> **Response**
>
> Thank you for your review!
>
> **Efficiency & Compute:**
> Thank you for this comment and discussion and we have added experiments to further evaluate the efficiency. We emphasize that our pre-training is a one-time cost to learn the "language" of the codec, and with current models, the inference cost is far greater than the training cost. It is hard to directly compare to other models, because models such as ViViT, InternVideo, etc. all start with an image-pretrained backbone, which itself takes various amounts of compute, depending on the backbones, pre-training datasets and tasks.  We have added an experiment to the paper comparing a ViT to our BytesSSM, using the same pre-training and compute setup and fixed training steps or wall-clock time. We acknowledge that there is more work to be done to further improve efficiency of training byte-based models. As this is the first work exploring video bytes and it is quite different from pixel-based understanding (we have no priors, such as the spatial prior provided by convolution in pixel-based models), the goal of the paper is to demonstrate that learning from bytes without any structure or knowledge of the input format is feasible for videos and even long videos, which can enable further research in this direction.
>
>
> **Long-Video QA evaluation experiments details:**
> We have updated the paper with more details about the Video QA evaluation experiments. Thank you for this suggestion. We follow the standard evaluation protocols and the results are comparable to reported approaches. Since visual question answering (VQA) requires running a text decoder to generate the output answer as text, we follow the previously established evaluation protocols for VQA with free-form text output, the same settings as Mirasol, LLaVA-OV-7B, etc. We have included more details in the manuscript.
> Specifically, for ActivityNet-QA, we output free-form text which, after normalization, is directly compared for an exact match to the ground truth. In terms of output normalization, we apply the standard stripping of whitespaces, lowercase normalization, articles and punctuation removal. If the output matches exactly the ground truth it is counted as correct, and incorrect otherwise. While this is a more challenging setting than a multiple-choice or classification setting (with limited options), we prefer it as it is more natural and has become the standard setting of other works. To our knowledge the compared prior approaches are evaluated by the same protocol. This dataset is fairly uncomplicated to evaluate as such since the answers are typically short.
> For CinePile, there are several answer choices per question, where additional distractor choices are provided in order to make the task more challenging. The answers are much longer sentences, but the challenge here is to select the most accurate and comprehensive answer from the multiple-choice options. The output here is a multiple choice answer, which is also fairly straightforward to evaluate and is directly comparable to the SOTA and is the standard evaluation method for current VLMs on CinePile.
>
> **‘Unlimited’ inference length claims:**
> We agree and we have revised the wording and have updated the manuscript with regards to ‘unlimited’ tokens at inference, e,g,, by removing the claim from the main claims and further noting that in practice it is constrained and provided more context.  We also updated the manuscript discussing that longer sequences at test times can yield better results as evidenced, but there are also practical limitations of very long-sequence length learning.Thank you for the suggestion.
>
> **Kinetics-400 and CinePile**
> We acknowledge the Reviewer’s comments and we note that we have included comparisons to much bigger or otherwise more powerful models. For example for CinePile, the better model is of about 7B parameters with additional image-based pre-training which has been improved over many generations of models. Similarly, the Kinetics-400 benchmark has been the subject of research for a very long time and some of the models we compared to are advantaged by more and more diverse training data or modalities, e.g. InternVideo, or for example stronger pre-training regimes, e.g. VideoMAE, ST-MAE, which can be further explored.

---

### Decision · Action_Editor_J7Hm · 2026-06-01

**Recommendation:** Accept as is

**Additional Comments:**

Three reviewers raised several substantive concerns in the initial round, including the strength of the efficiency claims, the lack of matched RGB baselines, incomplete QA evaluation details, the framing of the sequence-parallelism contribution, and the need for a more explicit discussion of the augmentation burden. The authors’ revision and responses have addressed these issues. All three reviewers give the Leaning Accept recommendation.

**Audience:**

Yes

**Audience Explanation:**

The authors claim that video understanding directly from encoded video byte streams is feasible. This is relevant to researchers working on video understanding, efficient machine learning, compressed-domain learning, and long-context sequence modeling.

**Claims And Evidence:**

Yes

**Claims Explanation:**

The submission’s main claim is supported by the evidence. The paper provides substantial empirical support across action recognition, long-video QA, high-FPS recognition, and audio-video understanding tasks. The architectural ablations, sequence-length scaling experiments, augmentation analysis, and codec/encoding robustness experiments make a convincing case that byte-level video modeling can work at scale.